# Effect of Metal Oxide Semiconductor Field-Effect Transistor Output Parasitic Capacitance on Efficiency in Full-Bridge LLC DC/DC Converters

**DOI:** 10.3390/mi15030309

**Published:** 2024-02-23

**Authors:** Ming-Hung Chen, Chia-Wen Hsieh

**Affiliations:** Department of Electrical Engineering, Ming Chi University of Technology, New Taipei City 243303, Taiwan; m08128014@mail2.mcut.edu.tw

**Keywords:** output parasitic capacitance (*C_oss_*), full-bridge LLC DC/DC converter, zero voltage switching (ZVS) state

## Abstract

This study analyzed the efficiency impact of a MOSFET output parasitic capacitance ($C_{oss}$) on a full-bridge LLC DC/DC converter. The core of the converter was the control chip for a half-bridge LLC DC/DC converter, and the output signal of the chip controlled the first-arm power transistors of the primary side of the converter. The coupling transformer reversed the output signal to control the primary side of the second arm of the power transistor. The full-bridge converter comprises a half-bridge control chip that converts the high-voltage DC power supply to a low-voltage DC power supply, which is then synchronously rectified and supplied to the load. The primary side of the power transistor achieves a zero-voltage switching (ZVS) state through the resonance of the LLC converter. This design gives the converter high power density and a simple structure. Furthermore, to determine the appropriate output parasitic capacitance for improving converter efficiency, this study analyzed the effect of the output parasitic capacitance on the switching loss and conduction loss of the power transistor on the basis of the output parasitic capacitance of the primary-side power transistor. A 1200 W converter prototype was fabricated in this study, and when the output was 300 W, efficiency increased from 92.603% to 93.462%, a 0.859% increase. The empirical results verified the feasibility of the proposed theory.

## 1. Introduction

Electric vehicles, cloud technology, and artificial intelligence are emerging industries whose development is hindered by power requirements. In particular, electric vehicles are more efficient at converting power from their power plant to vehicular motion relative to conventional combustion-engine vehicles—where the conversion efficiency of an electric vehicle is determined both by the motor and the overall power conversion system. In addition, cloud technology will become increasingly important as remote work becomes the norm. Powerful servers and high-power systems are required to handle large volumes of data being transmitted in and out of the cloud. These developments necessitate the design of a compact, high-conversion-efficiency, and high-power system [1].

DC/DC converters are either isolated or non-isolated depending on whether the converter has a transformer. Common non-isolated converters include the buck converter, boost converter, and buck-boost converter. These converters have simpler topologies and thus simpler circuitry; however, they cannot achieve high conversion efficiency when the step-down ratio is high due to the lack of a transformer and also cannot provide electrical isolation. Non-isolated DC/DC converters are more commonly found in buck-boost applications at the back end of isolated converters.

Common isolated converters include the flyback converter, half-bridge converter, and full-bridge converter. Isolated converters have transformers and therefore feature high step-down ratios and electrical isolation capability [2,3,4,5,6,7]. Flyback converters have simple structures and can be constructed with a few components; because these converters have only one power transistor, the voltage stress is rather high, and flyback converters are thus typically low-power converters. Half-bridge converters have one more power transistor than flyback converters and consequently lower voltage stress. Because the transformer inputs are half-wave signals, the operating current of the power transistor is higher than that of flyback converters. In applications with high input voltages and high power, choosing an appropriate power transistor is critical. Relative to half-bridge converters, full-bridge converters have two more power transistors, which reduce voltage stress, and use full-wave signals as transistor inputs, which further reduce the operating current of the power transistors while increasing control complexity. These converters are primarily used in high-voltage and high-power applications [8,9,10].

The LLC resonant converter selected in this study can be employed in both half-bridge and full-bridge architectures and uses pulse frequency modulation to maintain the duty cycle of the transistor at approximately 50% by modulating the driving signal to modulate the power output. The LLC resonant converter also has the following features [11,12,13]:In any state, the primary-side power transistor of the converter remains in the ZVS state; in a fully loaded state, the secondary-side rectifier enters the zero current switching state;When fully loaded, the switching frequency of the converter will be equal to the resonant frequency, improving its performance;The cutoff switching current of the power transistor can be lowered by increasing $L_{m}$, thus reducing switching loss;$L_{r}$ and $L_{m}$ can be integrated or isolated. Integrating the two can effectively reduce the size of the converter and save on costs, whereas isolating the two enables the precise control of the resonant inductance value and enhances converter efficiency.

Table 1 presents the basic characteristics of an LLC resonant transformer when used in a half-bridge converter or a full-bridge converter.

The full-bridge architecture is generally used in high-power and high-efficiency converters. However, given the lack of commercially available control chips for full-bridge LLC converters, in this paper, a full-bridge LLC resonant converter was fabricated using a half-bridge LLC resonant control chip. In order to maintain full-bridge power transistors in the ZVS states with a half-bridge LLC resonant control chip, the output parasitic capacitance of the first-arm power transistors was determined so that they can be added appropriately to improve the converter efficiency.

## 2. Operating Principles

Figure 1 illustrates the circuitry of the full-bridge LLC resonant converter featured in this study. The parameters labeled in the diagram are presented in Table 2. If the transformer in the converter were ideal, the primary side and secondary side would each form independent loops. The primary side is controlled by using the half-bridge LLC resonant control chip to drive the first-arm power transistors ($Q_{1}$ and $Q_{2}$), then reversing the first-arm signal through the coupling transformer to drive the second-arm power transistors ($Q_{3}$ and $Q_{4}$). The secondary side comprises a full-wave rectification circuit, which is equivalent to a diode ($D_{r1}$ and $D_{r2}$) in circuit analysis.

LLC resonant converters typically operate either in the inductive region or the resistive region, which are differentiated by the current mode of the converter. Figure 2 is a waveform diagram of the LLC resonant converter operating in the resistive region [14].

In Figure 2, the circuitry of the full-bridge LLC resonant converter has six modes, designated $t_{0}$ to $t_{6}$; the circuit operations are as follows [15,16,17]:Mode 1: ($t_{0}\leq t<t_{1}$)

The current path in this mode is depicted in Figure 3. At time $t_{0}$, $Q_{1}$ and $Q_{4}$ are turned on while $Q_{2}$ and $Q_{3}$ are cut off; in addition, $i_{L}$ flows through $Q_{1}$ and $Q_{4}$ and increases positively in a sinusoidal manner while $i_{m}$ increases linearly. At this time, because $i_{L}>i_{m}$, the primary side of the converter has a positive half-cycle voltage, and the energy is transmitted to the secondary side through the transformer. $D_{r1}$ is turned on and generates $i_{Dr1}$ to provide energy to $C_{out}$ and $R_{load}$. In this mode, $L_{m}$ is clamped by the output voltage to ${nV}_{o}$ and is not involved in the resonance; only $L_{r}$ and $C_{r}$ are involved.
Mode 2: ($t_{1}\leq t<t_{2}$)

The current path in this mode is depicted in Figure 4. During this interval, all four power transistors are cut off; because the inductor current cannot be cut off immediately, $i_{L}$ and $i_{m}$ remain equal, and their directions do not change. The converter makes use of this time interval to transfer energy from the output parasitic capacitor, charging $C_{oss1}$ and $C_{oss4}$ and discharging $C_{oss2}$ and $C_{oss3}$. During this time, the transformer is not transferring energy, and, consequently, $i_{Dr1}$ on the secondary side drops to zero; $R_{load}$ is supplied power by $C_{out}$. $L_{m}$ is no longer clamped by the output voltage and becomes part of the resonance alongside $L_{r}$ and $C_{r}$; at this time, $i_{L}$ can be regarded as a fixed current coming from the source.
Mode 3: ($t_{2}\leq t<t_{3}$)

The current path in this mode is depicted in Figure 5. During this interval, the four power transistors remain cut off. Again, because the inductor current cannot be cut off immediately, $i_{L}$ and $i_{m}$ remain equal, and their directions do not change. Furthermore, the charging and discharging output parasitic capacitors on the power transistor have been completed, and the remaining energy now flows through the body diode rather than the output parasitic capacitors as before; consequently, the current from the converter flows through $D_{SD2}$ and $D_{SD3}$ during this interval, and the transformer continues to have no energy to transfer. $R_{load}$ is still supplied power by $C_{out}$.
Mode 4: ($t_{3}\leq t<t_{4}$)

The current path in this mode is depicted in Figure 6. At time $t_{3}$, $Q_{2}$ and $Q_{3}$ are turned on, and $Q_{1}$ and $Q_{4}$ are cut off; $i_{L}$ passes through $Q_{2}$ and $Q_{3}$ and increases negatively in a sinusoidal manner, whereas $i_{m}$ decreases linearly. At this time, because $i_{L}<i_{m}$, the primary side of the converter has a negative half-cycle voltage, and the energy is transmitted to the secondary side through the transformer. $D_{r2}$ is turned on and generates $i_{Dr2}$ to provide energy to $C_{out}$ and $R_{load}$. In this mode, $L_{m}$ is clamped by the output voltage to ${nV}_{o}$ and is not involved in the resonance; only $L_{r}$ and $C_{r}$ are involved.
Mode 5: ($t_{4}\leq t<t_{5}$)

The current path in this mode is depicted in Figure 7. During this interval, the four power transistors are cut off; because the inductor current cannot be cut off immediately, $i_{L}$ and $i_{m}$ remain equal, and their directions do not change. Furthermore, the output parasitic capacitors of $Q_{2}$ and $Q_{3}$ are charged, and the output parasitic capacitors of $Q_{1}$ and $Q_{4}$ are discharged. During this time, the transformer is not transferring energy; consequently, $i_{Dr2}$ on the secondary side drops to zero, and $R_{load}$ is supplied power by $C_{out}$. $L_{m}$ is no longer clamped by the output voltage and becomes part of the resonance alongside $L_{r}$ and $C_{r}$; at this time, $i_{L}$ can be regarded as fixed current from the source.
Mode 6: ($t_{5}\leq t<t_{6}$)

The current path in this mode is depicted in Figure 8. During this interval, the four power transistors remain cut off. Again, because the inductor current cannot be cut off immediately, $i_{L}$ and $i_{m}$ remain equal, and their directions do not change. Furthermore, the charging and discharging output parasitic capacitors on the power transistor have been completed, and the remaining energy now flows through the body diode rather than the output parasitic capacitors as before; consequently, the current from the converter flows through $D_{SD1}$ and $D_{SD4}$ during this interval, and the transformer continues to have no energy to transfer. $R_{load}$ is still supplied power by $C_{out}$. At this point, the switching cycle is completed and starts again.

## 3. Effect of Output Parasitic Capacitor on Converter Efficiency

When the LLC resonant converter is under a full load range, the primary-side power transistor operates in the ZVS state. The converter resonates during the dead time to transfer the energy from $C_{OSS}$ of the power transistor (Figure 6). If the dead time is too short, the $C_{OSS}$ energy cannot be fully transferred, resulting in switching loss. In addition, energy has time to flow through the body diode of the power transistor if the dead time is overly long, increasing the conduction loss of the diode. Hence, the overall efficiency of the converter is affected by the amount of dead time. In practice, the dead time only lasts up until the conversion of $C_{OSS}$ energy is complete. Therefore, the half-bridge LLC resonant control chip employed in this paper adjusts the length of the dead time by detecting whether the $C_{OSS}$ energy has been released. The time needed to release the $C_{OSS}$ energy is determined by the current load, which means that each load has a corresponding $C_{OSS}$ energy release time, which is the corresponding dead time.

The amount of dead time affects the switching loss and conduction loss of a power transistor. When the system is under a heavy load, the power loss of the converter comes primarily from the iron loss and copper loss of the transformer, resulting in milder effects from switching loss and conduction loss of the power transistor. The effects of dead time under a light load were analyzed in this study. Light loads function in the inductive region, where the operating state is different from that in the resistive region—when the converter is in the resistive region, its current is in critical conduction mode, and, consequently, the transformer has no energy to transfer during the dead time. When the converter is in the inductive region, the current is in a continuous conduction mode, and the transformer continues to transfer energy during the dead time, as depicted in Figure 9; the equivalent circuit is shown in Figure 10.

Based on the direction of current, the converter stores energy in $C_{oss1}$ and $C_{oss4}$ and discharges energy from $C_{oss2}$ and $C_{oss3}$; if the energy transferred in the loop is fixed, the following can be obtained:(1)$$Q_{AB}=C_{oss}V_{AB}=i_{L}t_{ZVS}$$

Here, $V_{AB}$ and $Q_{AB}$ are the resonant tank voltage and total charge, and $C_{oss}$ is a pair of output parasitic capacitors (either $C_{oss1}$ and $C_{oss4}$ or $C_{oss2}$ and $C_{oss3}$); $t_{ZVS}$ is the dead time needed to reach ZVS. If the energy being transferred is fixed, in each state, only one pair of output parasitic capacitors is fully charged. According to Figure 9, transferring the loop energy to $C_{oss1}$ and $C_{oss4}$ at this time results in the following:(2)$$\left(C_{oss1}//C_{oss4}\right)V_{AB}=i_{L}t_{ZVS}$$

If the input voltage of the resonant tank is $+V_{in}$, then, according to Figure 10,
(3)$$\left(C_{oss1}//C_{oss4}\right)V_{in}=i_{L}t_{ZVS}$$
(4)$$i_{L}=i_{m}+i_{p}$$
(5)$$i_{L}=\frac{{nV}_{out}}{L_{m}}t_{{ZVS}^{2}}+\frac{I_{out}}{n}t_{ZVS}$$

In Equations (3) and (5), the time needed to reach ZVS in ideal conditions is as follows:(6)$$\left(C_{oss1}//C_{oss4}\right)V_{in}=\frac{{nV}_{out}}{L_{m}}t_{{ZVS}^{2}}+\frac{I_{out}}{n}t_{ZVS}$$
(7)$$\frac{{nV}_{out}}{L_{m}}t_{{ZVS}^{2}}+\frac{I_{out}}{n}t_{ZVS}-\left(C_{oss1}//C_{oss4}\right)V_{in}=0$$
(8)$$t_{ZVS}=\frac{-\frac{I_{out}}{n}\pm \sqrt{{\left(\frac{I_{out}}{n}\right)}^{2}+4\left(\frac{{nV}_{out}}{L_{m}}\right)\left(C_{oss1}//C_{oss4}\right)V_{in}}}{2\frac{{nV}_{out}}{L_{m}}}$$

A greater $I_{o}$ value corresponds to a heavier load and a shorter time required to reach ZVS. Therefore, when $I_{o}$ = 0 A, the longest ZVS time of the converter is $t_{ZVS}=115.576\;ns$. In this study, the efficiency of the converter under light loads was adjusted, and when the load was 300 W, ideally, $t_{ZVS}=26.641\;ns$.

The primary-side power transistors of the LLC resonant converter can enter the ZVS state under any load. The first-arm power transistors are driven by a half-bridge LLC resonant controller chip, and the second-arm power transistors are driven by the coupling transformer loop; as such, the controller chip is unable to detect the state of the second-arm power transistors. Furthermore, the coupling transformer loop causes signal delays, leading to errors in the driving times of the first-arm and second-arm power transistors; consequently, the second arm is unable to enter the ZVS state.

All four power transistors can enter the ZVS state by adjusting the first-arm $C_{oss}$ and extending the converter dead time. If any pair of output parasitic capacitors have the same amount of energy, then
(9)$$Q_{Coss1}=Q_{Coss4}$$
(10)$$\frac{i_{1}}{V_{Coss1}}t_{ZVS1}=\frac{i_{4}}{V_{Coss4}}t_{ZVS2}$$

Here, $V_{Coss1}$ and $V_{Coss4}$ are the voltages of the $Q_{1}$ and $Q_{4}$ output parasitic capacitors, respectively, and $t_{ZVS1}$ and $t_{ZVS2}$ are the dead times needed by the first and second arms, respectively. According to Kirchhoff’s circuit laws, the sum of the potential differences across all components in the loop is zero. Therefore, in the loop, $V_{Coss}=V_{in}$, yielding
(11)$$\frac{i_{L}}{V_{in}}t_{ZVS1}=\frac{i_{L}}{V_{in}}t_{ZVS2}$$

Due to the time differences between the first-arm and second-arm transistors, the second-arm power transistors were unable to discharge all of their energy during the dead time, indicating a need for more time. Hence, $t_{ZVS1}<t_{ZVS2}$, and by entering $t_{ZVS2}$ into Equation (6), we obtain:(12)$$\frac{{nV}_{o}}{L_{m}}t_{{ZVS2}^{2}}+\frac{I_{o}}{n}t_{ZVS2}-\left(C_{oss,New}\right)V_{in}=0$$
(13)$$C_{oss,New}=\frac{\frac{{nV}_{o}}{L_{m}}t_{{ZVS2}^{2}}+\frac{I_{o}}{n}t_{ZVS2}}{V_{in}}$$
(14)$$C_{oss1}+C_{ossX}=C_{oss,New}$$
where $C_{oss,New}$ is the adjusted output parasitic capacitance of the first-arm power transistors.

Using Equation (13), we find the appropriate first-arm $C_{oss}$ is 90.49 pF, and, using Equation (14), we determine that the first-arm power transistors must be connected in parallel with an output capacitor of 40.49 pF.

After the four primary-side power transistors have entered the ZVS state, if the first-arm $C_{oss}$ continues to be increased to extend the converter dead time, because the first-arm power transistors are monitored by the controller chip, the first arm remains in the ZVS state; however, the second-arm power transistors are not monitored, and, therefore, after the energy transfer is complete, the remaining energy of the second-arm $C_{oss}$ flows through the body diode of the power transistors, as illustrated in Figure 11; see Figure 12 for the equivalent circuit.

As illustrated in Figure 12, an excessive $C_{oss}$ results in overly long dead times; the conduction loss increased by the second-arm power transistors is
(15)$$P_{ZVS,Con}=f_{r}i_{L}V_{D_{SD3}}\left(t_{ZVS,exc}-t_{ZVS,mod}\right)$$
where
$t_{ZVS,exc}$: excessive time in the ZVS state$t_{ZVS,mod}$: optimal time in the ZVS state

Equation (14) indicates that when the dead time exceeds $t_{ZVS,mod}$, for each additional 10 ns, the conduction loss by the body diode increases by 8.2 μW.

The relationships between $C_{oss}$ and the switching loss and conduction loss of the primary-side power transistors of the full-bridge LLC resonant converter can be derived from the aforementioned equations (Figure 13).

The left (Region A) and right (Region B) partitions of the curve in Figure 13 correspond to values lower and higher than the suitable $C_{oss}$ value, respectively. Region A is the switching loss caused by the inability of the second-arm power transistors to fully transfer the $C_{oss}$ energy during the dead time, which is too short because $C_{oss}$ is too low. Region B is the conduction loss caused by the induction current continuing to flow though the body diode after the second-arm power transistors have finished transferring $C_{oss}$ because the dead time is too long. The figure demonstrates that loss increases when $C_{oss}$ is too high or too low. Furthermore, the loss in Region A is greater than the loss in Region B, and therefore, switching loss must be minimized in the design.

## 4. Experimental Results

The circuitry of the full-bridge LLC resonant converter combined with secondary-side synchronous rectification is depicted in Figure 14; the component specifications are presented in Table 3.

The converter’s poor efficiency under light loads was improved by adjusting the $C_{oss}$ of the power transistors. According to the calculation results of Equation (8), the first-arm power transistors on the primary side must be connected in parallel to a 40.94 pF capacitor to achieve the appropriate dead time; because the $C_{oss}$ cross voltage of each power transistor is $V_{in}$, a capacitor with a capacitance of 10pF/1kV was selected in this study. As indicated in Figure 15, overlap between the power transistors was greatly reduced, indicating noticeable ZVS states, and the measured temperatures of the two low-side power transistors was also greatly reduced. Prior to the adjustments, the second arm was 10 °C warmer than the first arm; after the adjustments, the temperature difference dropped to 0.5 °C.

According to the results of Equation (15), if the first-arm $C_{oss}$ is continually increased, the first arm remains in the ZVS state, while in the second arm, the remaining energy flowing through the body diode of the power transistors during the unnecessary dead time leads to conduction loss. As shown in Figure 16, after the transfer of the parasitic output capacitor energy, the power transistors failed to immediately change states.

.

When the converter is under a heavy load, the copper loss and iron loss of the transformer is greater than the switching loss and conduction loss of the power transistors. Furthermore, adjusting the $C_{oss}$ value does not significantly improve the efficiency. Hence, the $C_{oss}$ experiment in this paper was conducted under a light load, where components with $C_{oss}=10\;pF$ and 50 pF were connected parallel to each other; the observed efficiency changes are depicted in Figure 17. According to the empirical results, when the first-arm power transistors were connected in parallel to the component with $C_{oss}$ = 70 pF, relative to a parallel connection with an unadjusted parasitic capacitance of 50 pF, the efficiency for 300 and 600 W were higher (92.603%vs.93.462% for 300 W) and (95.193% vs. 95.302% for 600 W).

## 5. Conclusions

In this paper, a full-bridge LLC resonant converter was fabricated with a half-bridge LLC resonant controller chip that output higher power with the same input voltage, reducing the voltage stress on the power transistors and increasing the conversion efficiency. The first-arm power transistors were controlled with a controller chip, and the second-arm power transistors were driven with a coupling transformer loop. The converter dead time was extended by adjusting the $C_{oss}$ of the first-arm power transistors to ensure that all four power transistors entered a ZVS state, thereby increasing the efficiency of the converter under a light load. Synchronous rectification was adopted on the secondary side to enhance the overall converter efficiency. Thus, a 1200 W full-bridge LLC resonant converter was achieved. In this experiment, when the output was 300 W, the converter efficiency increased from 92.603% to 93.462% and the maximum efficiency of the converter circuits reached 96.762%.

## Figures and Tables

**Figure 1 micromachines-15-00309-f001:**
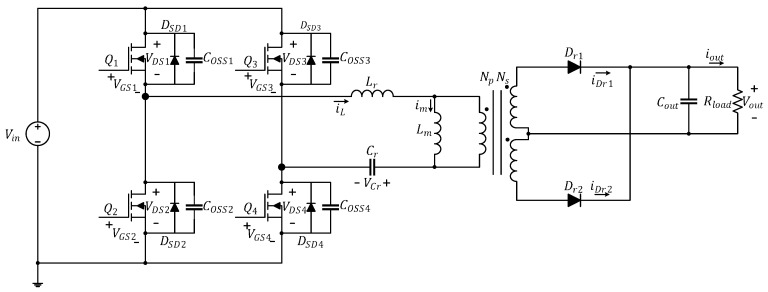
Circuitry of the full-bridge LLC resonant converter.

**Figure 2 micromachines-15-00309-f002:**
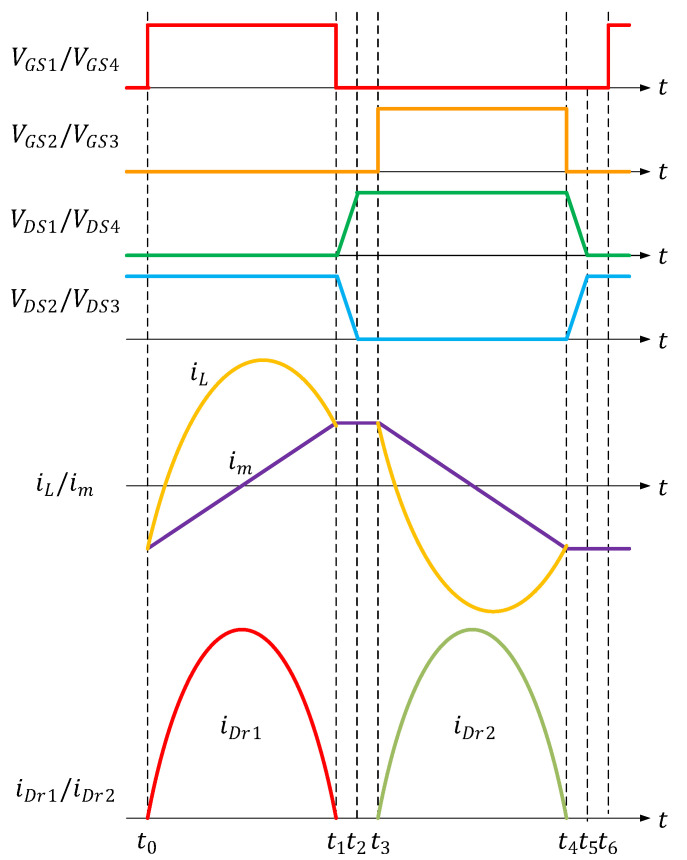
Waveform of a full-bridge LLC resonant converter in operation.

**Figure 3 micromachines-15-00309-f003:**
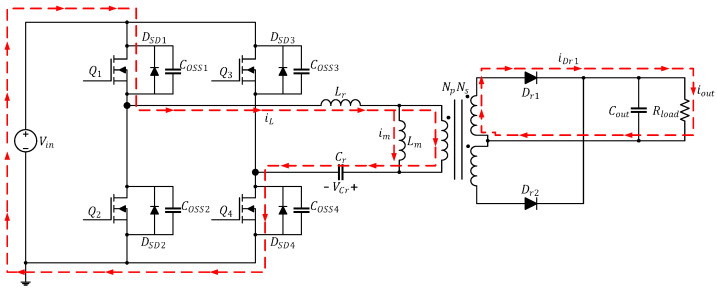
Current path of the full-bridge LLC resonant converter in Mode 1 ($t_{0}\leq t<t_{1}$).

**Figure 4 micromachines-15-00309-f004:**
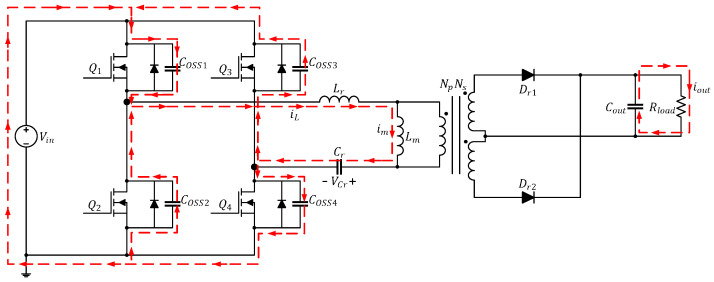
Current path of the full-bridge LLC resonant converter in Mode 2 ($t_{1}\leq t<t_{2}$).

**Figure 5 micromachines-15-00309-f005:**
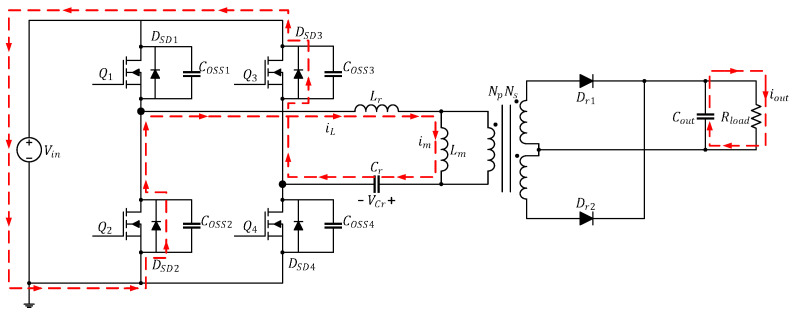
Current path of the full-bridge LLC resonant converter in Mode 3 ($t_{2}\leq t<t_{3}$).

**Figure 6 micromachines-15-00309-f006:**
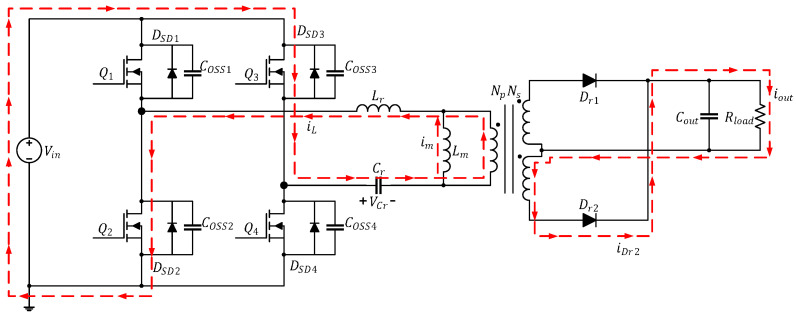
Current path of the full-bridge LLC resonant converter in Mode 4 ($t_{3}\leq t<t_{4}$).

**Figure 7 micromachines-15-00309-f007:**
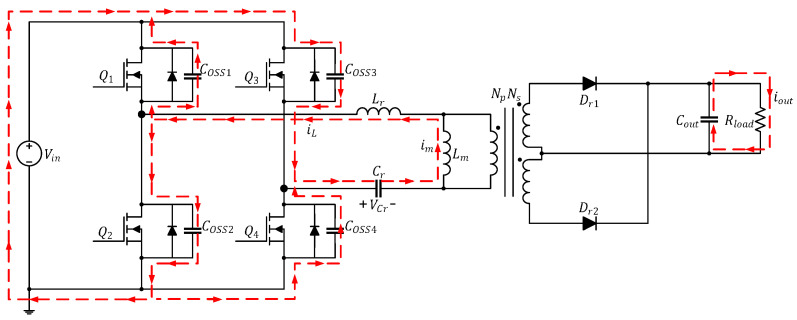
Current path of the full-bridge LLC resonant converter in Mode 5 ($t_{4}\leq t<t_{5}$).

**Figure 8 micromachines-15-00309-f008:**
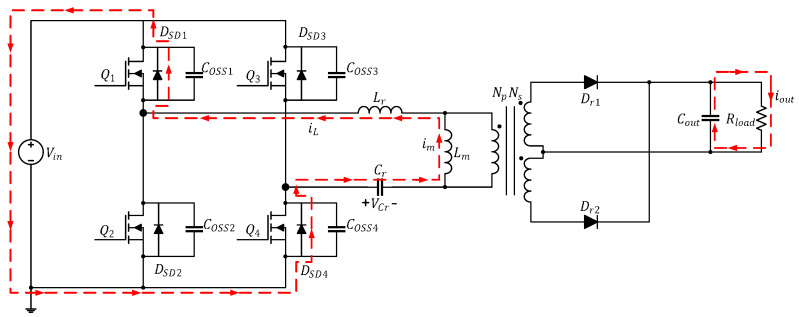
Current path of the full-bridge LLC resonant converter in Mode 5 ($t_{5}\leq t<t_{6}$).

**Figure 9 micromachines-15-00309-f009:**
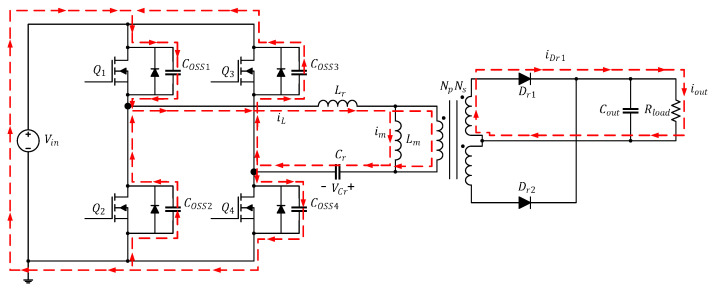
Full-bridge LLC resonant converter under a light load, Mode 2.

**Figure 10 micromachines-15-00309-f010:**
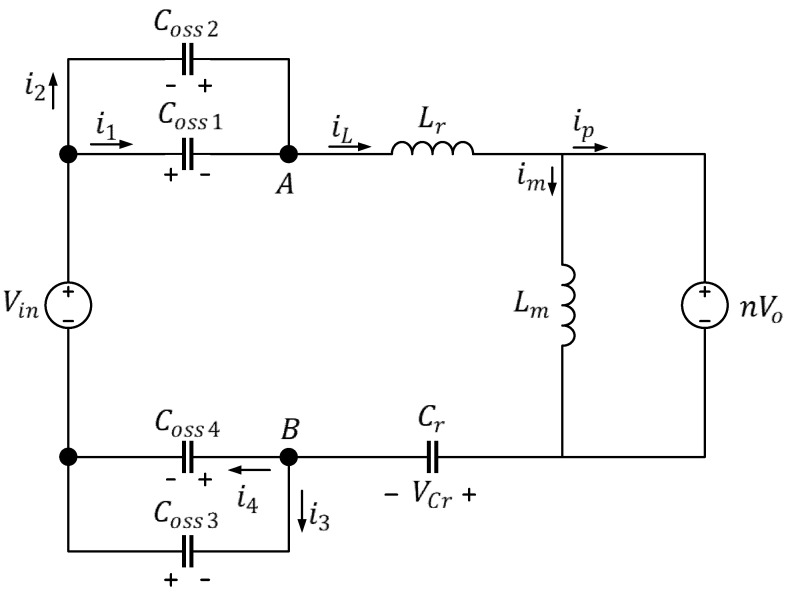
Equivalent circuit of the full-bridge LLC resonant converter under a light load, Mode 2.

**Figure 11 micromachines-15-00309-f011:**
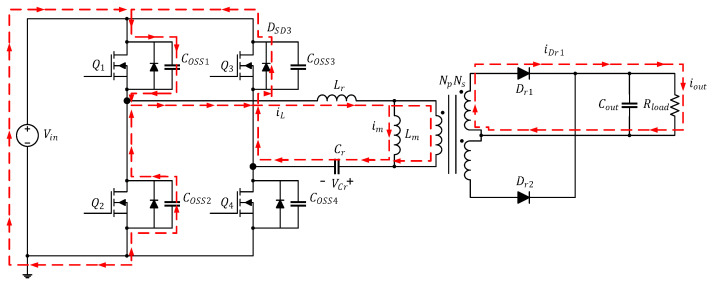
Full-bridge LLC resonant converter under light load and extended dead time in Mode 2.

**Figure 12 micromachines-15-00309-f012:**
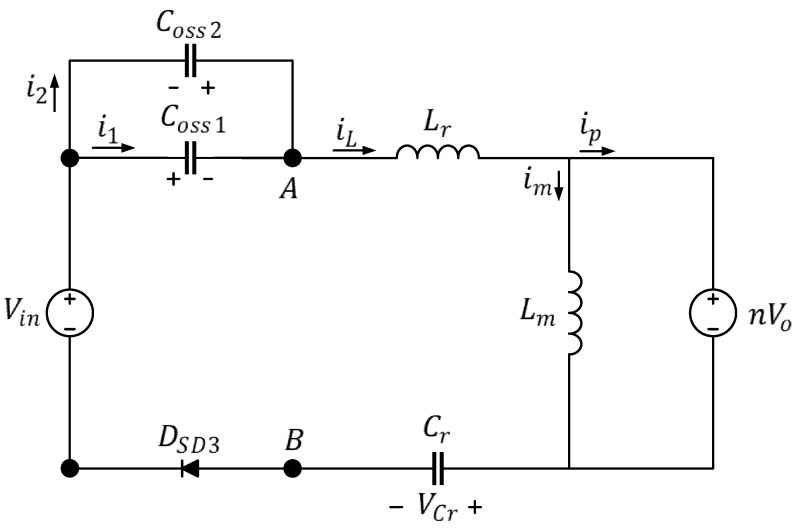
Equivalent circuit of the full-bridge LLC resonant converter under light load and extended dead time in Mode 2.

**Figure 13 micromachines-15-00309-f013:**
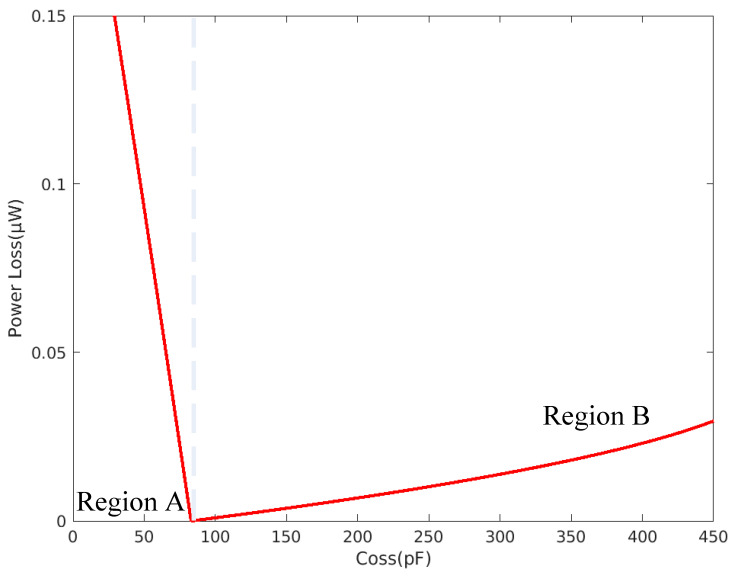
Effect of $C_{oss}$ on switching loss and conduction loss of the power transistors.

**Figure 14 micromachines-15-00309-f014:**
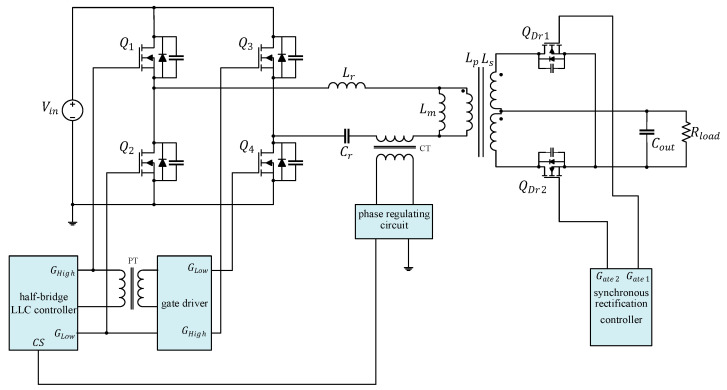
Complete schema of the full-bridge LLC resonant converter.

**Figure 15 micromachines-15-00309-f015:**
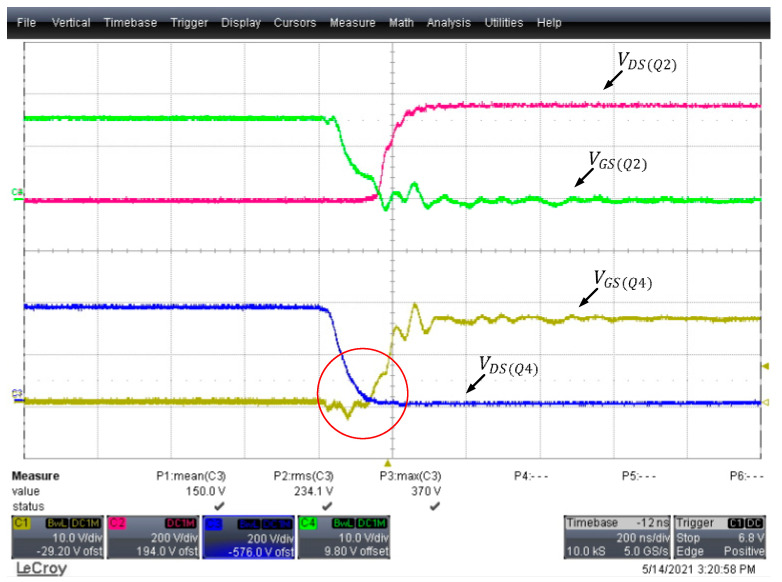
Dead time after adjusting the first-arm $C_{oss}$ of the full-bridge LLC resonant converter.

**Figure 16 micromachines-15-00309-f016:**
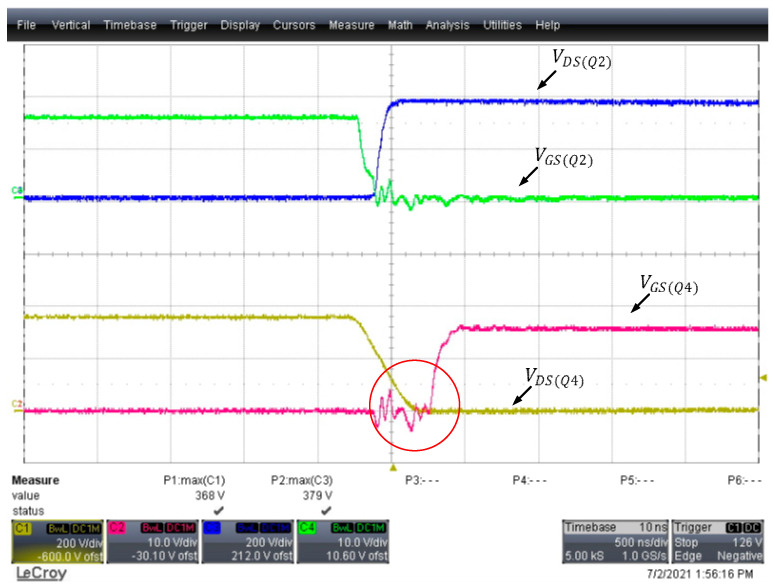
Parallel connection of the first-arm with excessive $C_{oss}$.

**Figure 17 micromachines-15-00309-f017:**
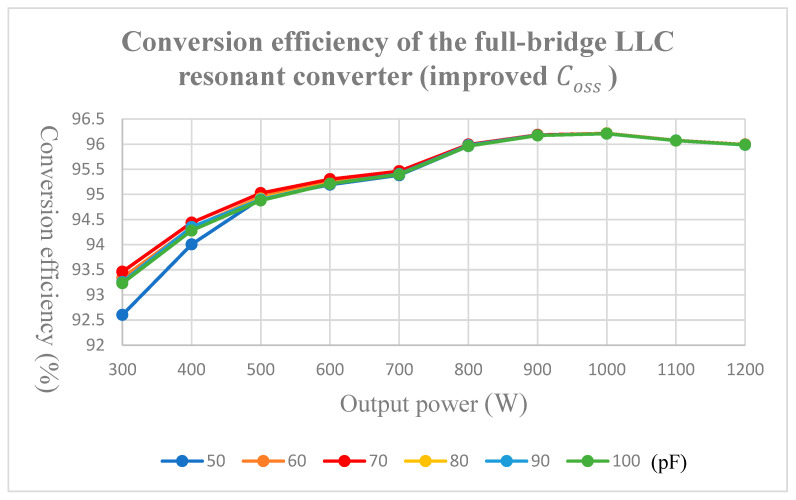
Conversion efficiency of the full-bridge LLC resonant converter (improved $C_{oss}$).

**Table 1 micromachines-15-00309-t001:** Half-bridge LLC resonant converter versus full-bridge LLC resonant converter.

Converter	Half-Bridge LLC Resonant Converter	Full-Bridge LLC Resonant Converter
Power transistors	2	4
Resonance slot reference point	${+V}_{in}$ and 0	${+V}_{in}$ $\;\mathrm{and}\;{-V}_{in}$
Control complexity	Simple	Complex
Applied power	Low power	High power
Control chips	Common	Does not exist

**Table 2 micromachines-15-00309-t002:** Circuit symbols.

Symbols	Names	Symbols	Names
$N_{p}$	Number of turns in the primary side	$N_{s}$	Number of turns in the secondary side
$Q_{1}$	First-arm high-side power transistor	$Q_{2}$	First-arm low-side power transistor
$Q_{3}$	Second-arm high-side power transistor	$Q_{4}$	Second-arm low-side power transistor
$C_{oss1}$	$Q_{1}$ output parasitic capacitor	$C_{oss2}$	$Q_{2}$ output parasitic capacitor
$C_{oss3}$	$Q_{3}$ output parasitic capacitor	$C_{oss4}$	$Q_{4}$ output parasitic capacitor
$D_{SD(1\sim 4)}$	Body diode of the power transistor	$C_{r}$	Resonant capacitor
$L_{r}$	Resonant inductor	$L_{m}$	Magnetizing inductor
$D_{r1}$	Positive half-cycle rectifier diode on the secondary side	$D_{r2}$	Negative half-cycle rectifier diode on the secondary side
$C_{out}$	Output filter capacitor	$R_{load}$	Secondary-side load resistor
$V_{in}$	Input voltage	$V_{GS(1\sim 4)}$	Power transistor gate-source voltage
$V_{Cr}$	Resonant capacitor voltage	$V_{DS(1\sim 4)}$	Power transistor drain-source voltage
$V_{out}$	Output voltage	$i_{out}$	Output current
$i_{L}$	Resonant inductor current	$i_{m}$	Magnetizing inductor current
$i_{Dr1}$	$D_{r1}$ forward current	$i_{Dr2}$	$D_{r2}$ forward current

**Table 3 micromachines-15-00309-t003:** Converter specifications.

Specifications	Values
$\mathrm{Input}\;\mathrm{voltage}\;\mathrm{range}\;\left(V_{in}\;Range\right)$	$350\;V_{dc}\sim 420\;V_{dc}$
$\mathrm{Output}\;\mathrm{voltage}\;\left(V_{out}\right)$	$48\;V_{dc}$
$\mathrm{Resonant}\;\mathrm{frequency}\;\left(f_{r}\right)$	82 kHz
$\mathrm{Resonant}\;\mathrm{capacitance}\;\left(C_{r}\right)$	66 nF
$\mathrm{Resonant}\;\mathrm{induction}\;\left(L_{r}\right)$	60 μH
$\mathrm{Magnetizing}\;\mathrm{inductance}\;\left(L_{m}\right)$	520 μH
Primary-side LLC resonant controller chip	NCP13992
Secondary-side synchronous rectification controller chip	NCP4318

## Data Availability

Data are contained within the article.

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
