# Peer review of "Effect of Metal Oxide Semiconductor Field-Effect Transistor Output Parasitic Capacitance on Efficiency in Full-Bridge LLC DC/DC Converters"

_micromachines, 2024, doi:10.3390/mi15030309_

Round 1

Reviewer 1 Report

Comments and Suggestions for Authors

The authors analyzed the impact of MOSFET output parasitic capacitances on a self-designed full-bridge LLC DC/DC converter. It is surprising that the authors claimed that there is no full bridge controller commercially available. A quick google search found a list of such controller chips, for example, https://www.infineon.com/cms/en/product/power/gate-driver-ics/full-bridge-drivers/; https://octopart.com/pulse/p/full-bridge-mosfet-driver-selection-and-design-guide; https://www.ti.com/lit/ds/symlink/lm5045.pdf?ts=1707727121011&ref_url=https%253A%252F%252Fwww.google.com%252F;

The authors need to re-evaluate the research background, compare their solutions with the other ones both commercially and academically available. And after this, the novelties and advantages in their solutions should be highlighted, elaborated, and benchmarked against the other ones.      

Reviewer 2 Report

Comments and Suggestions for Authors

Here are some comments:

1. Is the output Parasitic Capacitance, the only factor we need to consider? how about Parasitic inductor or Parasitic Capacitance at other locations?

2. In Figure2, the current curve looks a bit strange, iL rising and decreasing like an arc? iDr1 and iDr2 have the same issue? The im cant be a straight line I think

3. What is the difference between: Resonant capacitance in Table 3 and Parasitic Capacitance?  Please give more deep explanations

4. Fig.17, overlapping for pF.

Round 2

Reviewer 1 Report

Comments and Suggestions for Authors

The authors have addressed my comments properly.

Reviewer 2 Report

Comments and Suggestions for Authors

The Figure2 still looks weird . I think the author should redraw this figures.